# Formulation, Characterization, Anti-Inflammatory and Cytotoxicity Study of Sesamol-Laden Nanosponges

**DOI:** 10.3390/nano12234211

**Published:** 2022-11-26

**Authors:** Anroop B. Nair, Pooja Dalal, Varsha Kadian, Sunil Kumar, Archana Kapoor, Minakshi Garg, Rekha Rao, Bandar Aldhubiab, Nagaraja Sreeharsha, Rashed M. Almuqbil, Mahesh Attimarad, Heba S. Elsewedy, Pottathil Shinu

**Affiliations:** 1Department of Pharmaceutical Sciences, College of Clinical Pharmacy, King Faisal University, Al Hofuf 31982, Saudi Arabia; 2Department of Pharmaceutical Sciences, Guru Jambheshwar University of Science and Technology, Hisar 125001, India; 3Atam Institute of Pharmacy, Om Sterling Global University, Hisar 125001, India; 4School of Pharmaceutical Sciences, Delhi Pharmaceutical Sciences and Research University, New Delhi 110017, India; 5Department of Pharmaceutics, Vidya Siri College of Pharmacy, Off Sarjapura Road, Bangalore 560035, India; 6Department of Pharmaceutical Sciences, College of Pharmacy, AlMaarefa University, Dariyah, Riyadh 13713, Saudi Arabia; 7Department of Biomedical Sciences, College of Clinical Pharmacy, King Faisal University, Al Hofuf 31982, Saudi Arabia

**Keywords:** *Sesamum indicum*, phenolic bioactive, melanoma, nanosponges, anti-inflammatory, cell viability

## Abstract

Sesamol (SES) possesses remarkable chemotherapeutic activity, owing to its anti-inflammatory and antioxidant potential. However, the activity of SES is mainly hampered by its poor physicochemical properties and stability issues. Hence, to improve the efficacy of this natural anti-inflammatory and cytotoxic agent, it was loaded into β-cyclodextrin nanosponges (NS) prepared using different molar ratios of polymer and crosslinker (diphenyl carbonate). The particle size of SES-laden NS (SES-NS) was shown to be in the nano range (200 to 500 nm), with a low polydispersity index, an adequate charge (−17 to −26 mV), and a high payload. Field emission scanning electron microscopy, thermogravimetric analysis, and Fourier transform infrared spectroscopy were used to characterize the bioactive-loaded selected batch (SES-NS6). This batch of nanoformulations showed improved solubilization efficacy (701.88 µg/mL) in comparison to bare SES (244.36 µg/mL), polymer (β-CD) (261.43 µg/mL), and other fabricated batches. The drug release data displayed the controlled release behavior of SES from NS. The findings of the egg albumin denaturation assay revealed the enhanced anti-inflammatory potential of SES-NS as compared to bare SES. Further, the cytotoxicity assay showed that SES-NS was more effective against B16F12 melanoma cell lines than the bioactive alone. The findings of this assay demonstrated a reduction in the IC_50_ values of SES-NS (67.38 μg/mL) in comparison to SES (106 μg/mL). The present investigation demonstrated the in vitro controlled release pattern and the enhanced anti-inflammatory and cytotoxic activity of SES-NS, suggesting its potential as a promising drug delivery carrier for topical delivery.

## 1. Introduction

As the largest organ of the human body, the skin functions as the main protective wall against numerous assaults from the external environment, which includes ultraviolet radiation, pathogens, and physical and chemical fluctuations [1]. Skin cancer activates the excessive growth of cells in the outermost layer, i.e., the epidermis, initiated by unrepaired DNA impairment in cells, resulting in mutations. Such modified epidermal cells propagate rapidly and generate malignant tumors [2]. In this way, abnormal dermis layers synchronize with the physiological alterations and prompt abnormal environments, leading to different types of cancer. Skin cancer/melanoma is the most prevalent form of cancer, having one million new patients annually [3]. Some well-known treatment modalities for this cancer include surgical excision, photodynamic therapy, radiation, cryotherapy, and chemotherapy. However, the available conventional treatments have also been accompanied by severe toxicity, pain, intense inflammation, and unpleasant scars among patients, which result in poor compliance. Another challenge with these treatments includes acute adverse effects, which interrupt the patient’s immune system. To address these issues, the use of natural phytoconstituents in skin cancer management could be a safe, affordable, and efficient alternative. With respect to this, researchers have taken a keen interest in the pharmacopeia of traditional medicines [4]. A sufficient number of properties, including anti-inflammatory, antioxidant, and immunomodulatory effects, are considered responsible for their possible chemopreventive actions, resulting in the suppression or reversal of carcinogenesis [5].

In general, natural products possess a high molecular weight, which in turn limits their skin permeation and requires additional enhancement approaches. Recently, Zeinali et al. reported the prevention of skin cancer by using a nanoethosome formulation loaded with Gamma Oryzanol [6]. Naturally available anti-cancer bioactives include vinca alkaloids, podophyllotoxin and its derivatives, taxanes and their analogs, anthracyclines, camptothecin and its derivatives, and many more. Plant-based products or their derivatives make up half of the authorized anti-cancer molecules that are currently accessible [7]. Sesamol (SES) is one such natural phenolic compound, extracted from the seeds and oil of sesame (*Sesamum indicum*). SES has been widely studied for its therapeutic activities and demonstrated its potential as a regulator of metabolism, owing to its antioxidant, anti-hepatotoxic, anti-mutagenic, anti-aging, anti-inflammatory, and chemopreventive activities. This bioactive has a broad range of biological activities responsible for its anti-cancer potential. These include lipid peroxidation inhibition and boosted free radical scavenging, antioxidant enzyme upregulation, suppression of interleukin-1β and TNFα (tumor necrosis factor) expression, inhibition of NF-κB (nuclear factor kappa B) signaling, concealment of 5-lipoxygenase (5-LOX) and oxidized low-density lipoprotein receptor 1 (LOX-1) expression, stimulation of apoptosis, cell growth arrest, and the modulation of p53, caspase-3, BCL-2, and Bax expression [8]. Notwithstanding these interesting features, its clinical application is hampered owing to its stability issues. The favorable physicochemical properties of SES, such as solubility, log P octanol/ water, and molecular weight, i.e., 38.8 ± 1.2 mg/mL, 1.29 ± 0.01, and 138.34 g/mol, respectively, indicate it to be a fascinating and distinctive bioactive. This bioactive shows poor stability in response to temperature and light. The adequate log P value aids the easy permeability of SES via the dermal route, but the same feature is also expected to cause its speedy transport via all the skin layers, ultimately reaching the systemic circulation, followed by metabolism and clearance. However, all these issues restrict SES’s local effects on the skin tissue [9]. To deal with this problem, there is an urgent need to fabricate an innovative carrier system for SES. Some investigations have been performed to improve the efficacy of this bioactive by scientists using various nanocarriers, such as nanostructure lipid carriers [9], carbohydrate polymers [10], floating beads [11], emulsions [12], micelles [13], and gelatin nanoparticles [14]. Previously, our research group encapsulated SES into nanosponges (NS) and successfully enhanced the stability of this bioactive [15]. These promising results encouraged us to explore SES-NS further for topical delivery.

Nanomedicine has been evidenced to enhance the probability of the implementation of both synthetic and plant-based moieties by upgrading their activity, boosting their sustained release, reducing the dose, and elevating their pharmacological actions [16,17]. NS is a nano-sized porous carrier system fabricated by the hyper-interlinking of polymers. These carrier systems hold the ability to improve the solubility, penetration, absorption, bioavailability, physicochemical stability, sustained and targeted delivery, and therapeutic potential of numerous bioactives. Further, our research team has previously explored this delivery system for various bioactives, including babchi oil, azelaic acid, ellagic acid, and p-coumaric acid [18,19,20]. The NS-based drug carriers are a promising and profound research domain, predominantly to address the aforementioned obstacles associated with the current anti-cancer formulations and improve cancer therapy [21].

Different anti-cancer drugs, such as resveratrol [22], doxorubicin [23], paclitaxel [24], quercetin [25], curcumin [26], tamoxifen [27], and camptothecin [28], have been successfully encased in NS and demonstrated modulated antitumor effects [23]. Further, this nanosystem can be employed for targeting skin cancer by ameliorating the drug’s potential to reach and penetrate the cancer site [29]. Keeping this in view, NS as a carrier is a promising concept and paves the way for the control and management of skin cancer. Further, having a basis for this delivery system and given the encouraging results of previous delivery systems, we sought to encase SES into NS to evaluate its anti-cancer and anti-inflammatory activity. Thus, the current research aimed to explore NS as a potential delivery system for encasing SES to offer an innovative, non-invasive clinical treatment option for melanoma. The SES-NS was developed and subsequently screened from physicochemical and technological viewpoints (i.e., mean particle size, zeta potential, size distribution, encapsulation efficacy, and in vitro release). In addition, the anti-inflammatory activity and cytotoxicity against B16F12 melanoma cell lines were investigated. Overall, SES-NS seems to be an attractive option to explore in order obtain an effective treatment for skin cancer. Furthermore, integration with a suitable topical delivery vehicle, such as creams, gels, and emulgels, may profoundly enhance the efficacy of this chosen moiety in skin cancer applications.

## 2. Materials and Methods

### 2.1. Materials

SES was obtained from Sigma Chemical Co., St Louis, MO, USA. Roquette, Lestrem, France supplied the β-Cyclodextrin (β-CD). Sigma Aldrich, Bangalore, India, provided the diphenyl carbonate (DPC). All other chemicals and reagents were of analytical quality and were used without being further purified.

### 2.2. Methods

#### 2.2.1. Preparation and Purification of NS

Five types of NS were formulated with varying molar ratios of polymer (β-CD) to crosslinker (DPC) (1:2, 1:4, 1:6, 1:8, and 1:10). Briefly, predetermined amounts of both polymer and DPC were allowed to react at 90 °C for 5 h. Once the crosslinking was done, the reaction product was cooled down at normal-temperature conditions, and to exclude unreacted β-CD, an additional quantity of double-distilled water was used. Thereafter, the resultant washed product was extracted with a Soxhlet apparatus using acetone for up to 4 h to eliminate impurities and byproducts. Then, the nanoformulations were vacuum-dried [30].

#### 2.2.2. Solubilization Efficiency of NS

To measure the improvement in solubilization, the solubility of the SES in polymer and all the fabricated NS variants was determined. A surplus of the SES was suspended together with a predetermined amount (20 mg) of NS in water (20 mL). The suspension was allowed to shake for 24 h at an ambient temperature. The obtained suspensions were centrifuged to separate the supernatant and bare drug. Thereafter, supernatants were filtered and scanned using a UV spectrophotometer at λ_max_ 294 nm [31].

#### 2.2.3. Preparation of SES-NS

SES was encapsulated into NS by employing the freeze-drying technique, as established earlier [32]. An aqueous solution of individual NS and the required amount of SES was added into various batches of formulations. The obtained suspension was sonicated (10 min) and stirred at ambient temperature for 24 h. After centrifuging the solution (at 3000 rpm for 15 min), the uncomplexed SES was settled as a residue below the colloidal supernatant, and the obtained supernatant was freeze-dried. After lyophilization, different formulation batches (SES-NS2, SES-NS4, SES-NS6, SES-NS8, and SES-NS10) were collected and stored.

#### 2.2.4. Estimation of Encapsulation Efficiency

A fixed amount of NS was suspended in methanol and sonicated (10 min) to segregate the uncomplexed SES. The obtained solution was analyzed via spectrophotometry at λ_max_ 294 nm against a blank [33]. The encapsulation efficiency of SES in NS was computed using the following equation:(1)$$\%\;\mathrm{Encapsulation}\;\mathrm{Efficiency}=\frac{\mathrm{Weight}\;\mathrm{of}\;\mathrm{sesamol}\;\mathrm{loaded}\;\mathrm{nanosponges}}{\mathrm{Weight}\;\mathrm{of}\;\mathrm{initially}\;\mathrm{used}\;\mathrm{sesamol}}\times 100$$

#### 2.2.5. Characterization of Prepared SES-NS

All fabricated batches of SES-NS were characterized for size and polydispersity indices by employing a Malvern Zetasizer (Malvern Instruments Ltd., Worcestershire, UK). A weighed amount of NS (10 mg) was dispersed in water (10 mL) and sonicated. Zeta potential calculation was carried out by employing an additional electrode in the same instrument. For the surface charge analysis, NS solutions were shifted into a capillary cell having platinum electrodes [34]. All of the readings were taken at 25.0 ± 1.0 °C.

#### 2.2.6. Fourier Transform Infrared Spectroscopy

Fourier transform infrared spectroscopy of bare SES, SES-laden NS, and NS was recorded from 4000 to 400 cm^−1^ via the Perkin Elmer Spectrum BX II (Waltham, MA, USA), to ensure the complex formation. The KBr pellet technique was employed for the analysis of samples [35].

#### 2.2.7. Thermal Analysis

Bare SES, SES-laden NS, and NS were analyzed for thermal study using the thermal analyzer TA-60WS (DSC-60 Plus, Shimadzu, Japan). A 10 mg sample was fixed in a standard aluminum pan and heated to temperatures between 30 °C and 350 °C (at a rate of 10 °C/minute). For reference, a standard blank pan was employed. During the analysis, the selected samples were subjected to nitrogen (flow rate of 30 mL/minute) to maintain the inert environment during the experiment [36].

#### 2.2.8. Field Emission Scanning Electron Microscopy

A field emission scanning electron microscope (FE-SEM, Hitachi Instruments (SU-8010) was used to check the surface morphology of fabricated NS and SES-NS. An aluminum stub having a diameter of 12.5 mm was attached to a double-sided coated carbon tab and on its surface samples were placed. The same samples were gold-coated before observation under an instrument at an acceleration voltage [37].

#### 2.2.9. In Vitro Release Profile of SES-NS

The release of SES and SES-NS was performed using the previously reported method, with some modifications [10,38]. An appropriate amount of SES and SES-NS was added into phosphate buffer solutions (pH 7.4 and pH 4.5) and added to a dialysis bag (cellulose membrane, molecular weight cut-off: 14,000 Dalton, pore size: 0.4 nm, Himedia). The dialysis membrane bags were immersed in 50 mL buffer solution and stirred at 100 rpm at body temperature. At predetermined intervals, an aliquot of sample was taken out for analysis and afterward restored with the same portion of fresh buffer. The released amount of SES was quantified by measuring the absorbance using a UV spectrophotometer (at λ_max_ 294 nm) [39].

#### 2.2.10. Albumin Denaturation-Based Anti-Inflammatory Activity

The anti-inflammatory activity of SES and a selected batch of SES-NS was evaluated with an in vitro assay of the heat-induced inhibition of protein denaturation, as described previously by Tunit et al. [40]. SES and SES-NS were dissolved in water to obtain various concentrations. The sample solutions (2 mL) were mixed with egg albumin (0.2 mL) and PBS 7.4 (2.8 mL). The control solution contained distilled water (2 mL) instead of the test solution. The solution was incubated (37 ± 2 °C for 15 min) and then heated (70 ± 2 °C for 5 min). The samples were cooled down and then analyzed at an absorbance of 660 nm by employing a UV–visible spectrophotometer [40]. The % inhibition of protein denaturation was obtained as follows:(2)$$\mathrm{Inhibition}\;\left(\%\right)=\frac{V_{1}}{V_{0}-V_{1}}\times 100$$
where *V*_1_ is the test sample absorbance and *V*_0_ is the absorbance of the control.

#### 2.2.11. Cytotoxicity Assay against B16F12 Melanoma Cell Line

The B16F12 cell lines were procured from the National Centre for Cell Science, Pune, India. To keep the cells alive, Dulbecco’s modified Eagle’s medium was used. A high concentration of glucose with fetal bovine serum (10%), along with an antibiotic–antimycotic solution (1%), in a CO_2_ (5%), O_2_ (18–20%) atmosphere, at a fixed temperature (37 °C) in the CO_2_ incubator was maintained, and it was sub-cultured every 2 days [41].

#### 2.2.12. Cell Viability Assay

B16F12 melanoma cells were seeded in a 96-well plate at a density of 20,000 cells/well and grown for 24 h. Thereafter, different concentrations of SES and SES-NS (10, 25, 50, 75, 100, and 200 µg/mL) were added and incubated for 48 h at a temperature of 37 °C in a CO_2_ atmosphere (5%). Afterward, we added MTT (3-(4,5-dimethylthiazol-2-yl)-2,5-diphenyl tetrazolium bromide) reagent to achieve a 0.5 mg/mL concentration and further incubated it for 3 h. Finally, the MTT reagent was removed, and we then added 100 μL of solubilization solution, and absorbance was measured at 630 nm with an ELISA reader [42]. The % cell viability was determined using the below formula:(3)$$\%\;\mathrm{cell}\;\mathrm{viability}=\left(\frac{\mathrm{Mean}\;\mathrm{abs}\;\mathrm{of}\;\mathrm{treated}\;\mathrm{cells}}{\mathrm{Mean}\;\mathrm{abs}\;\mathrm{of}\;\mathrm{Untreated}\;\mathrm{cells}}\right)\times 100$$

The IC_50_ value was calculated by employing a linear regression equation (y = mx + c). Herein, y = 50, and the values of m and c were calculated from the viability graph.

#### 2.2.13. Statistical Analysis

All the experimental data are expressed as the standard deviation (mean ± SD). Statistical comparisons were made by applying ANOVA. Herein, a statistically significant difference was shown as a *p*-value of <0.05.

## 3. Results and Discussion

In the present experiment, the melting technique was utilized for the fabrication of all batches of blank NS using different molar concentrations of DPC and β-CD, and their physicochemical analysis is summarized in Table 1.

The molar ratios of both the polymer and crosslinker employed in the fabrication of NS impact their practical yield (Table 1). It has been found that an enhancement in the molar ratio leads to an increase in the practical yield. The increased yield may be owing to the increased reactive functional groups at greater concentrations.

### 3.1. Solubility of NS

The solubility of bare SES, SES in polymer, and all fabricated batches of NS in distilled water is presented in Figure 1. The solubility of this bioactive with different NS batches was found to be higher than that of both bare SES (244.36 μg/mL) and SES with β-CD (261.43 μg/mL). The results of this study demonstrated that the solubility of SES was increased up to 2.37- and 2.9-fold with SES-NS4 and SES-NS6, respectively. However, the solubility was found to decrease with SES-NS8 and SES-NS10, owing to the higher concentration of DPC, which resulted in the formation of large nanochannels. These nanochannels in turn might have led to the leakage and release of the entrapped bioactive, as described elsewhere [27]. The results of the solubilization efficiency of all batches revealed significance. As the maximum solubility of SES was exhibited with SES-NS6 (Figure 1), this batch was selected for further characterization and evaluation studies.

### 3.2. Estimation of Encapsulation Efficiency

In order to check the impact of the β-CD:DPC ratio on % encapsulation efficiency, SES-NS were fabricated with variable ratios of both. The percentage of SES encapsulation in all batches of NS is presented in Table 2 and Figure 2. Among all crafted nanoformulations, %EE was remarkably higher for SES-NS6, followed by SES-NS4, SES-NS2, SES-NS8, and SES-NS10. Such outcomes revealed that the higher encapsulation of the drug into NS does not involve the usage of a higher amount of crosslinker, but rather an optimal quantity of crosslinker. As observed in SES-NS2, incomplete crosslinking occurs, owing to the lesser amount of crosslinker, which results in the formation of insufficient nano-cavities and, consequently, leads to a decrease in available sites for SES entrapment. Furthermore, using a greater quantity of crosslinker resulted in the fabrication of large and complex nano-networks. These networks may be responsible for the expulsion of the drug from these large cavities, as mentioned in previous studies [34]. The findings herein are also found consistent with previous results obtained in a study by Anandam et al. [25].

### 3.3. Characterization of Prepared SES-NS

The particle size, zeta potential, and polydispersity index of SES-NS were evaluated by using Milli-Q water, and the findings are reported in Table 2 and Figure 3. The size of SES-laden NS was found to be between 200 and 500 nm, with an acceptable range of polydispersity index values (between 0.347 ± 0.069 and 0.540 ± 0.099) [43]. Further, the zeta potential of all formulation batches was determined and the surface charges for these nanoparticles were found to be between −17.733 ± 0.577 and −26.603 ± 1.521 mV. The higher zeta values were attributed to a strong repulsive force between the NS, which resulted in good stability and prevented their accumulation [44].

### 3.4. Fourier Transform Infrared Spectroscopy

Figure 4 displays the Fourier transform infrared spectroscopy results of blank NS, bare SES, and SES-NS. Peaks of absorption were visible in the spectra of pure SES at 3400, 2916, 1490, 1268, 1188, 1036, and 836 cm^−1^, corresponding to the phenolic group, saturated -CH, out-of-plane CH bending, C-O of phenolic OH, aromatic C-H in-plane bending, symmetric stretching of =C-O-C, and -CH of adjacent hydrogens, respectively, according to the Fourier transform infrared spectrum of SES previously reported in the literature [15,45,46]. Our data are found to be consistent with these reports and hence confirm the purity of this bioactive. Because of the carbonyl group, blank NS showed a distinctive peak at around 1643 cm^−1^. As per literature reports, this peak represents a peculiar feature of NS and validated their formation [47]. Additionally, other prominent peaks of plain NS have also been observed, ranging from 3100 to 3500 cm^−1^, owing to O-H stretching vibration, and at 1434 cm^−1^ due to the bending and angular deformation of the C-H bond. Other peaks owing to the stretching vibration of di-substituted double bonds of =C-H and the bending vibration of tri-substituted double bonds of =C-H were seen at 894 cm^−1^ and 766 cm^−1^, respectively [25,48,49]. However, the SES-NS6 spectrum showed that the distinctive peaks of SES had either been widened or displaced, suggesting that the SES had been successfully encapsulated in NS.

### 3.5. Thermal Analysis

The DSC thermograms of NS, bare SES, and SES-NS6 are presented in Figure 5. The DSC thermogram of SES demonstrated a characteristic endothermic peak at 65 °C, indicating a bioactive melting point, whereas NS showed peaks at 75 °C, corresponding to the crystallization temperature of β-CD. In addition, no peak was obtained before 320 °C, indicating the stable and versatile nature of NS. This temperature is far beyond the storage conditions in a pharmaceutical setup. Further, the nonexistence of the SES characteristic peak in SES-NS6 suggested that an inclusion complex had formed [19].

### 3.6. Field Emission Scanning Electron Microscopy

The morphology of NS and SES-NS6 was investigated by FE-SEM studies (Figure 6). The images illustrated that the blank NS formulation had a solid, rough surface with a porous structure, wherein the bioactive molecules could be encased. Unlike these, SES-NS6 revealed filled pores, resulting in smooth surfaces. This might be accredited to the encapsulation of the bioactive into the porous structures of the nanosponges. Similar morphological images and results have also been reported previously [50].

### 3.7. In Vitro Release of SES-NS

Figure 7 shows the in vitro release patterns of bare SES and SES-NS (at pH 7.4 and 4.5). The release of bare SES was found to be 99% at pH 7.4 in 2 h and at pH 4.5 in a half-hour, respectively. SES-NS6 demonstrated the controlled and slow release of the bioactive, i.e., 62% and 100% in 24 and 48 h, respectively, at PBS (pH 7.4), whereas it showed 100% release at pH 4.5 (4 h). At acidic pH, SES exhibited faster release than at physiological pH. Besides the slow and sustained release of SES, the release pattern displayed no initial burst effect, which may be attributed to the inclusion of the complexation of bioactive into the nanostructures and also its crosslinking with NS. Moreover, it could also be suggested that there was no weakly absorbed or uncomplexed SES on the outside of the NS, and the bioactive might have been complexed strongly with NS and, thus, prevented its initial burst release, as described earlier [51]. The sluggish release behavior of encaged moieties from manufactured NS has also been noted by other researchers [20,26].

### 3.8. Albumin Denaturation-Based Anti-Inflammatory Activity

The anti-denaturation effect of bioactive and SES-NS was evaluated to assess their potential to reduce inflammation. The results are summarized in Figure 8, which shows that SES and SES-NS6 resulted in the dose-dependent inhibition of protein denaturation, i.e., 20.55 ± 0.78% and 32.40 ± 3.8% at 50 µg/mL, respectively. Denaturation was reduced by 70.70 ± 0.18% and 83.89 ± 0.96 at a concentration of 500 µg/mL for both SES and SES-NS6, respectively. Hence, from the findings, it was concluded that the SES nanoformulation showed higher anti-inflammatory potential than bare SES. Further, the anti-denaturation effect of this bioactive was observed at all concentrations; however, it increased at higher concentrations. In addition, as per literature reports, protein denaturation has been associated with inflammatory responses, resulting in inflammatory disorders [52]. Therefore, the reduction in denaturation by SES-NS6 in the present case may help in addressing inflammation and associated disorders such as skin cancer.

### 3.9. Cell Viability Assay

Evaluation of the cytotoxicity of the developed SES-NS is of vital importance for their possible use against skin cancer. For this, the effect of bare SES and SES-NS6 on the viability of B16F12 melanoma cells was explored by employing an MTT assay. It was noticed that the application of various chosen concentrations of SES and SES-NS6 on melanoma cells exerted inhibitory action on the cell viability/proliferation in a time- and concentration-dependent manner (Figure 9). The high cytotoxicity of SES (bare/encapsulated bioactive) can be ascribed to the anti-inflammatory [53], antioxidant [54], and anti-proliferative activity [55] of SES, as reported in previous studies on various tumor cells. These outcomes demonstrated that NS alone does not result in any cytotoxicity to cancer cells, suggesting the biocompatibility as well as the significance of this nanocarrier. It was demonstrated in the literature that moieties having cell viability higher than 80% are generally accepted as biocompatible. Moreover, melanoma cells (B16F12), after treatment with SES and SES-NS6, displayed a concentration-dependent cytotoxic pattern. The proliferation of the chosen melanoma cells was found to be dramatically diminished after the application of both SES and SES-NS6. However, when compared to SES, SES-NS6 remarkably decreased the cell viability of melanoma cells at all tested concentrations. This could be accredited to the sustained release behavior of the bioactive from NS, resulting in its continuous exposure to cancerous cells, generating its superior anti-cancer effect [56]. The IC_50_ values of bioactive SES and SES-NS6 were found at 106 and 67.38 μg/mL, respectively.

## 4. Conclusions

In the present study, host–guest complexes of SES and NS were successfully designed and crafted. Among all the prepared nanosponge batches, SES-NS6, which showed the highest solubilization and encapsulation capacity, was selected and characterized. Further, an evaluation of the release profile, anti-inflammatory activity, and cell viability was conducted. The findings indicated an augmentation in the aqueous solubility of SES after its encapsulation in NS, and higher encapsulation efficiency was displayed by SES-NS6. The subsequent experiments also showed that NS improved the anti-inflammatory potential of SES and its cytotoxic effect of SES against B16F12 melanoma cell lines when compared to the bare bioactive. Indeed, it is relevant to mention the attributes that make SES-NS a superior nanoformulation over other choices. Indeed, the simplistic process, low cost of the polymer as well as crosslinker, and absence of catalysts’ chemical alteration make this an attractive drug delivery carrier. Above all, the omission of the use of organic solvents provides an advantage to this nano-delivery system over its counterparts. According to the outcomes of the present work, the controlled release behavior of SES-NS6, assisted by its superior cytotoxicity to melanoma cells, along with the aforementioned merits, can represent a promising choice for the management of skin cancer. Furthermore, future research could focus on the in vivo evaluation of SES-NS6 by integrating it into a suitable topical delivery system to validate its potential for skin cancer.

## Figures and Tables

**Figure 1 nanomaterials-12-04211-f001:**
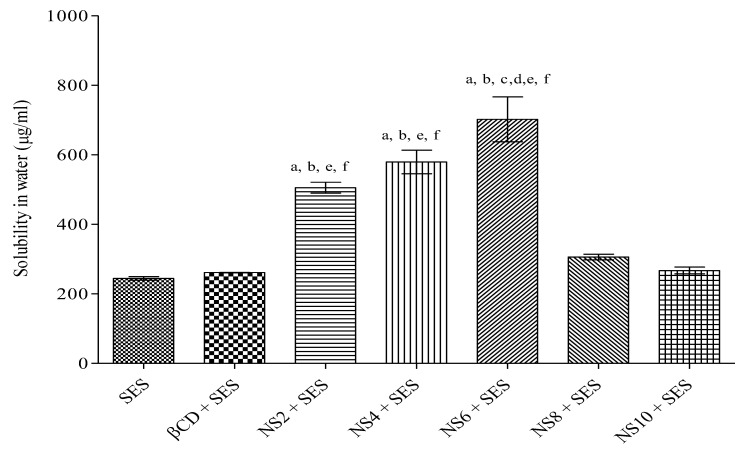
Solubilization of bare SES and SES along with β-CD, and various nanosponges (NS2, NS4, NS6, NS8, and NS10). Data represented as mean ± SD (n = 6). Statistical data were evaluated by using one-way ANOVA and subsequently using Tukey’s test for multiple data evaluations. (a) *p* < 0.001 vs. SES; (b) *p* < 0.001 vs. β-CD; (c) *p* < 0.001 vs. NS2; (d) *p* < 0.001 vs. NS4; (e) *p* < 0.001 vs. NS8; (f) *p* < 0.001 vs. NS10. SES: Sesamol; SES-NS: Sesamol-loaded nanosponge; β-CD: β-Cyclodextrin.

**Figure 2 nanomaterials-12-04211-f002:**
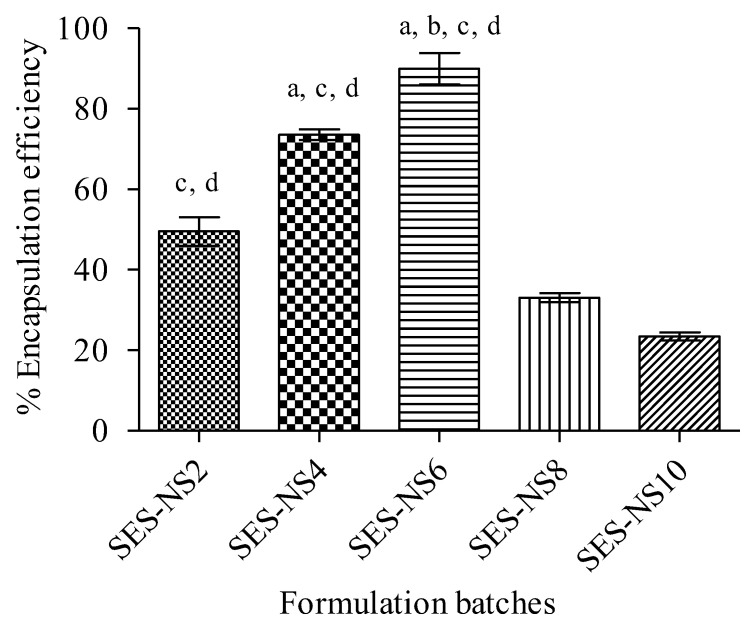
Encapsulation efficiency of SES–NS batches (1:2-1:10). Data reported here represent mean ± SD (n = 6). A one-way ANOVA statistical data analysis was followed by a Tukey’s test for multiple comparisons. (a) *p* < 0.001 vs. SES-NS2; (b) *p* < 0.001 vs. SES-NS4; (c) *p* < 0.001 vs. SES-NS8; (d) *p* < 0.001 vs. SES-NS10. SES: Sesamol; SES-NS: Sesamol-laden NS.

**Figure 3 nanomaterials-12-04211-f003:**
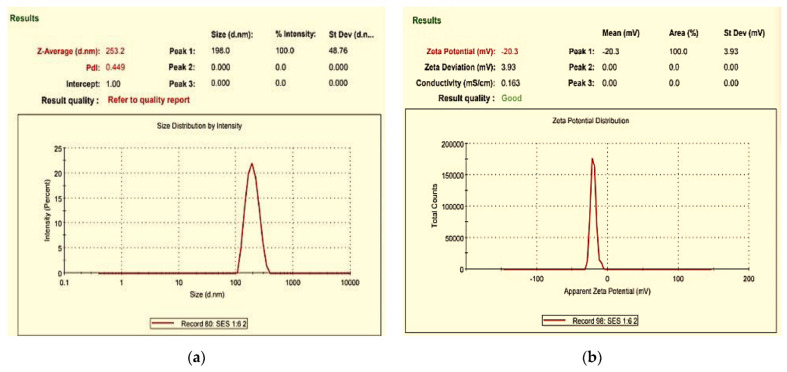
Representative images of (**a**) particle size and (**b**) zeta potential measurement of SES-NS.

**Figure 4 nanomaterials-12-04211-f004:**
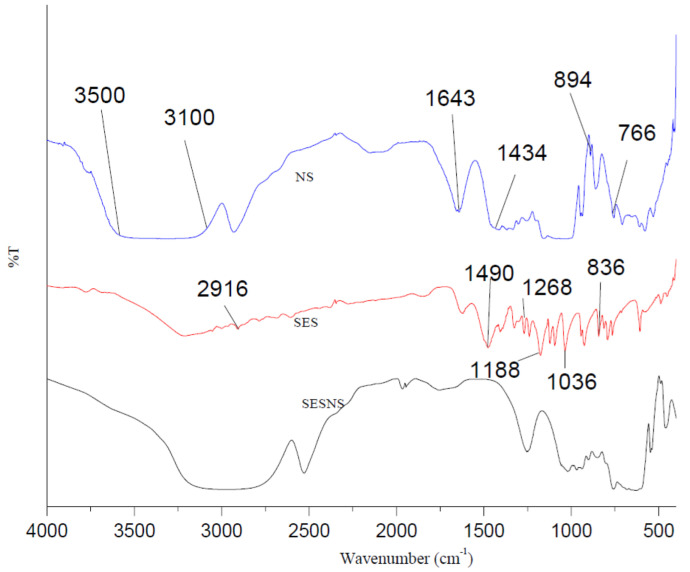
Fourier transform infrared spectra of blank nanosponges (NS), Sesamol (SES), and bioactive-loaded selected batch (SESNS).

**Figure 5 nanomaterials-12-04211-f005:**
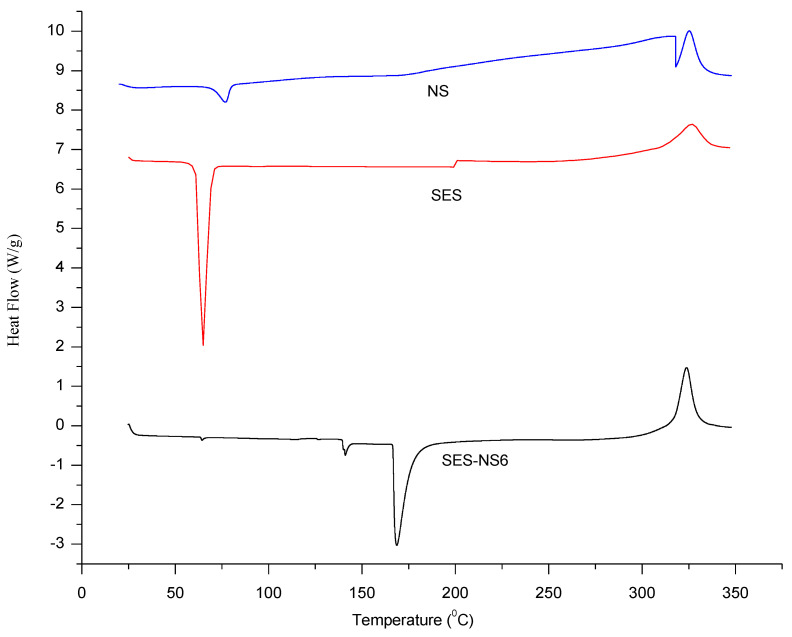
Differential scanning calorimetry thermograms of blank nanosponges (NS), sesamol (SES), and bioactive-loaded selected batch (SES-NS6).

**Figure 6 nanomaterials-12-04211-f006:**
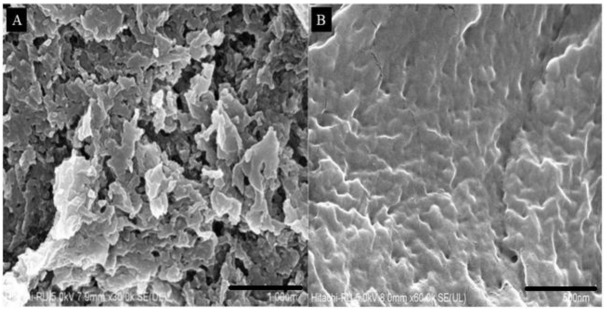
Field emission scanning electron microscopy images of (**A**) blank nanosponges (NS), and (**B**) bioactive-loaded selected batch (SES-NS6).

**Figure 7 nanomaterials-12-04211-f007:**
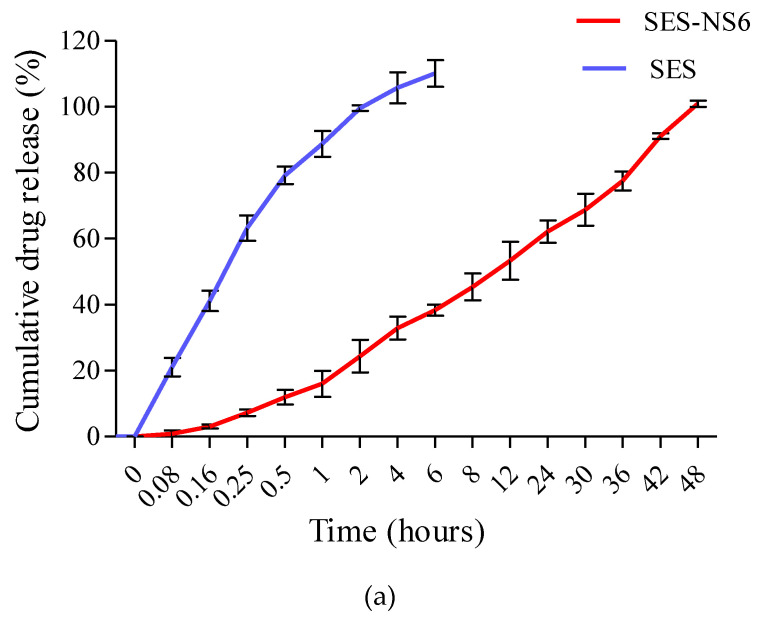
In vitro drug release profile of bare sesamol (SES) and sesamol-laden nanosponges (SES-NS6) in buffer solution at (**a**) pH 7.4 and (**b**) pH 4.5.

**Figure 8 nanomaterials-12-04211-f008:**
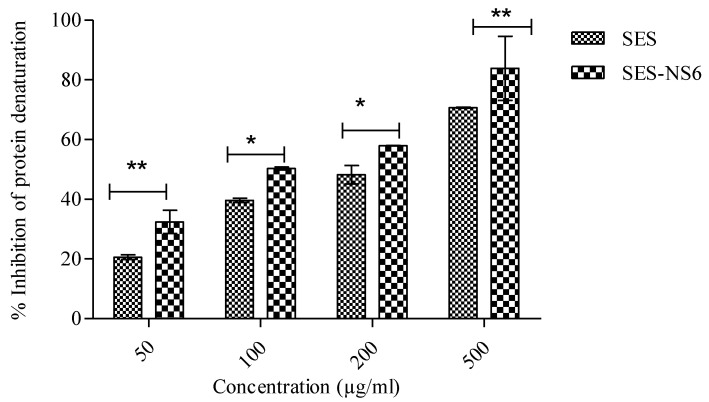
Percentage inhibition of protein denaturation by sesamol (SES) and sesamol-laden nanosponges (SES-NS6). Results are displayed as mean ± SD (n = 6). A two-way ANOVA statistical data analysis was followed by multiple comparisons by Bonferroni post-tests. Statistical significance ** indicates *p* < 0.001 and * indicates *p* < 0.05.

**Figure 9 nanomaterials-12-04211-f009:**
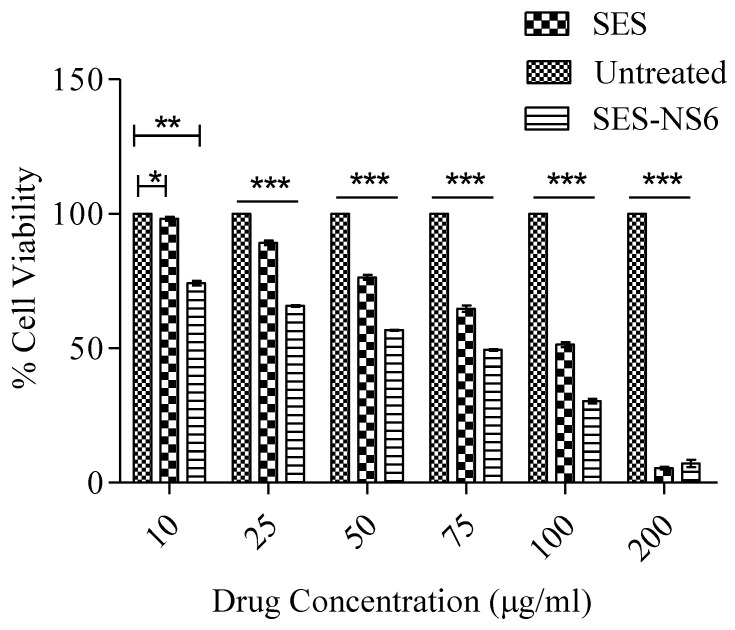
Results of MTT assay for cell viability after 48 h of incubation of B16F12 melanoma cell lines with native sesamol (SES) and sesamol-laden nanosponges (SES-NS6). Results are expressed as mean ±SD (n = 6). A two-way ANOVA statistical data analysis was followed by multiple comparisons by Bonferroni post-tests. Statistical interferences: statistically significant difference among untreated vs. SES *p* < 0.05 and untreated vs. SES-NS6 *p* < 0.001 at concentration 10 µg/mL; and statistically significant difference of SES-NS6 compared to the untreated group, SES, and SES-NS6 (*p* < 0.001) at other concentrations. Statistical significance *** shows *p* < 0.0001; ** shows *p* < 0.001, and * shows *p* < 0.05.

**Table 1 nanomaterials-12-04211-t001:** Composition of prepared nanosponges.

Nanosponges	Molar Ratio of β-CD: DPC	Quantity of β-CD (g)	Quantity of DPC (g)	Practical Yield (g)
SES-NS2	1:2	2.274	0.856	1.8956
SES-NS4	1:4	2.274	1.712	2.4689
SES-NS6	1:6	2.274	2.568	2.9845
SES-NS8	1:8	2.274	3.424	3.1278
SES-NS10	1:10	2.274	4.28	3.3720

β-Cyclodextrin (β-CD), diphenyl carbonate (DPC).

**Table 2 nanomaterials-12-04211-t002:** Physical properties of the aqueous dispersion of sesamol (SES) and sesamol-loaded nanosponge (SES-NS) batches.

Formulation Code	Encapsulation Efficiency ± SD (%)	Particle Size ± SD (nm)	Zeta Potential ± SD (mV)	Polydispersity Index ± SD
SES-NS2	49.508 ± 3.555	230.366 ± 30.050	−17.733 ± 0.0577	0.347 ± 0.069
SES-NS4	73.547 ± 1.362	266.033 ± 21.450	−22.233 ± 1.050	0.540 ± 0.099
SES-NS6	89.946 ± 3.939	251.533 ± 49.321	−20.333 ± 0.4509	0.444 ± 0.112
SES-NS8	33.055 ± 1.136	426.921 ± 68.608	−19.701 ± 0.710	0.356 ± 0.055
SES-NS10	23.493 ± 1.000	272.311 ± 45.696	−26.603 ± 1.521	0.455 ± 0.052

(Data presented here are mean ± SD, *n* = 6).

## Data Availability

The data presented in this study are contained within the article.

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
