# Peer review of "Formulation, Characterization, Anti-Inflammatory and Cytotoxicity Study of Sesamol-Laden Nanosponges"

_nanomaterials, 2022, doi:10.3390/nano12234211_

Round 1

Reviewer 1 Report

The manuscript by Nair et al. untitled "Formulation, Characterization, Anti-inflammatory and Cytotoxicity Study of Sesamol Laden Nanosponges Against Skin Cancer Cell Line" presents a set of experiments focused on preparation and characterization of SES-loaded nanosponges and their potential application. The study seems to be rather preliminary and needs further experiments and some corrections to make it suitable for publication.

Major remarks:

1/ Materials and Methods, lines 159-160 - the Authors claim that after centrifugation "the unreacted SES was removed". How is it possible to precipitate SES at 3000rpm for 15 min? Please explain the methodology to obtain the supernantant without unreacted SES?

2/ Materials and Methods, lines 202-203 - released SES was quantified spectrophotometrically at 294 nm, what is the base for such measurement? Is it selective for SES only?

3/ Results, p. 9/10 - how did the Authors conclude on SES encapsulation based on SEM pictures? Are there any methods to visualize the cargo on the NS surface?

4/ Results, Figure 7 - why the drug release profile was not continued to 100% release? Even if the drug was not released completely, the effects of SES-NS on cells viability were tested after 48h incubation, so the presented profile should be prolonged to 48h

5/ Results, p. 11 - lines 360-361 - is it possible that "denaturation was exceedingly elevated to 70.70%"? Or inhibited by 70%? Please, explain clearly the link between inhibited denaturation and anti-inflammatory potential. How can the proteins denature under physiological conditions?

6/ Results, p. 11 - lines 389 and further - the IC50 value itself gives no information about anti-cancer potential. The Authors should compare IC50 values for different cells (normal and cancerous) and then conclude on activity against skin cancer (if any).

7/ Results, p. 11 - lines 392-394 - the Authors assume that nanosponges have "high-solubilizing capacity" and modulate the membrane integrity, while in line 380 suggest the biocompatibility of this nanostructure. Please, explain this ambiguous statement.

Author Response

Response to Reviewer 1 Comments

The manuscript by Nair et al. untitled "Formulation, Characterization, Anti-inflammatory and Cytotoxicity Study of Sesamol Laden Nanosponges Against Skin Cancer Cell Line" presents a set of experiments focused on preparation and characterization of SES-loaded nanosponges and their potential application. The study seems to be rather preliminary and needs further experiments and some corrections to make it suitable for publication.

Response: We would like to convey our sincere thanks for the time and suggestions by the potential reviewer, which would surely improvise the quality of the article.

Point 1: Materials and Methods, lines 159-160 - the Authors claim that after centrifugation "the unreacted SES was removed". How is it possible to precipitate SES at 3000rpm for 15 min? Please explain the methodology to obtain the supernatant without unreacted SES?

Response: We appreciate the keen observation made by the reviewer. The comment has been addressed and the sentence has been rewritten in the manuscript (Lines 159-160). During centrifugation, it was noted that the uncomplexed drug gets settled as a residue below the colloidal supernatant. This has also been reported in the literature.

References supporting the procedure followed:

  1. Swaminathan, S., Vavia, P. R., Trotta, F., Cavalli, R., Tumbiolo, S., Bertinetti, L., & Coluccia, S. (2013). Structural evidence of differential forms of nanosponges of beta-cyclodextrin and its effect on solubilization of a model drug. Journal of inclusion phenomena and macrocyclic chemistry76(1), 201-211.
  2. Anandam, S., & Selvamuthukumar, S. (2014). Fabrication of cyclodextrin nanosponges for quercetin delivery: physicochemical characterization, photostability, and antioxidant effects. Journal of materials science49(23), 8140-8153.
  3. Swaminathan, S., Pastero, L., Serpe, L., Trotta, F., Vavia, P., Aquilano, D., ... & Cavalli, R. (2010). Cyclodextrin-based nanosponges encapsulating camptothecin: physicochemical characterization, stability and cytotoxicity. European Journal of Pharmaceutics and Biopharmaceutics74(2), 193-201.

Point 2:  Materials and Methods, lines 202-203 - released SES was quantified spectrophotometrically at 294 nm, what is the base for such measurement? Is it selective for SES only?

Response: We thank the reviewer for raising this question. The release of SES was quantified at 294 nm because it is the absorbance maxima for this bioactive as presented in the figure below. In addition, a standard curve was also plotted at this λmax in phosphate buffer (pH 7.4) and the R2 value was calculated. All these measurements (n=3) were base for the selection of this absorbance and are selective for sesamol.

Figure 1: Absorbance graph of sesamol (please see in attached pdf).

Figure 2: Standard curve of sesamol in PBS 7.4 pH (please see in attached pdf).

Point 3: Results, p. 9/10 - how did the Authors conclude on SES encapsulation based on SEM pictures? Are there any methods to visualize the cargo on the NS surface?

Response: We thank the reviewer for this comment. The blank nanosponge images showed small cavities whereas these cavities were filled with sesamol and a smooth surface was observed in SES-loaded NS images, which supported the encapsulation results. This has also been added to the manuscript as per the reviewer’s suggestion (Lines 335-338). Similar findings have been reported by Rao et al; 2013.

Rao, M., Bajaj, A., Khole, I., Munjapara, G., & Trotta, F. (2013). In vitro and in vivo evaluation of β-cyclodextrin-based nanosponges of telmisartan. Journal of inclusion phenomena and macrocyclic chemistry, 77(1), 135-145.

Point 4: Results, Figure 7 - why the drug release profile was not continued to 100% release? Even if the drug was not released completely, the effects of SES-NS on cells viability were tested after 48h incubation, so the presented profile should be prolonged to 48h.

Response: We appreciate the keen observation made by the reviewer. The drug release profile up to 100% release has been missed and updated now in the manuscript (Lines 343-347). This has also been incorporated in the graph (Figure 7B). In the case of SES-NS, the release was found to be 60% in 24 hours. As we have not continued release further, hence, the cell viability assay was extended up to 48 hours to evaluate the effect of sesamol released from NS in this period.

Point 5: Results, p. 11 - lines 360-361 - is it possible that "denaturation was exceedingly elevated to 70.70%"? Or inhibited by 70%? Please, explain clearly the link between inhibited denaturation and anti-inflammatory potential. How can the proteins denature under physiological conditions?

Response: We are sorry for this error as we overlooked it. Herein, denaturation was inhibited and in the mentioned sentence it was mistaken (lines 362-363). This has been addressed and corrected in the manuscript as per the reviewer’s suggestion.

Point 6:  Results, p. 11 - lines 389 and further - the IC50 value itself gives no information about anti-cancer potential. The Authors should compare IC50 values for different cells (normal and cancerous) and then conclude on activity against skin cancer (if any).

Response: We thank the reviewer for this suggestion. Yes, the suggestions by the reviewer are valuable and worth considering. As we have outsourced the cell line study, it is difficult for us to incorporate these suggestions in this work. However, we are working on skin cancer and consider these for our future studies.

Point 7: Results, p. 11 - lines 392-394 - the Authors assume that nanosponges have "high-solubilizing capacity" and modulate the membrane integrity, while in line 380 suggest the biocompatibility of this nanostructure. Please, explain this ambiguous statement.

Response: We are sorry for the confusion. The comment has been addressed according to the reviewer’s suggestion and the said sentence is now deleted in the revised text as it was confusing and the reviewer has raised concerns.

Reviewer 2 Report

There are many experiment that are missing that could prove the purposed hypothesis. So, this paper cannot accepted in its present form. However, the major concerns are:

1. SEM image is not clear. The author should provide good TEM and SEM image of nanosponge. 

2. To have good polydispersity index, the PDI should be below 0.3. However, the author showed the PDI higher than 0.4 (Figure 3A). Thus, the nanoparticle developed here doesn't have narrow size distribution.

3. Since, the nanosponge has cytotoxic effect in the cancer model, the author should perform the release study in the acidic pH too.

4. Major in vitro experiment like intracellular uptake study of nanoparticle is missing.

5. The author should be consistent with the terminology B16F10, somewhere it's mentioned B16F12 as well.

6. Animal experiment is encouraged to confirm its effect.

Author Response

Response to Reviewer 2 Comments

There are many experiment that are missing that could prove the purposed hypothesis. So, this paper cannot accepted in its present form. However, the major concerns are:

Response: We would also like to thank you for raising the concerns, which would improvise the quality of our manuscript.

Point 1. SEM image is not clear. The author should provide good TEM and SEM image of nanosponge. 

Response: We are sorry for this error. The SEM image has now been replaced with a clearer version of the image (Figure 6). However, we have not performed TEM for these samples. Only FE-SEM was carried out by employing Hitachi Instruments (SU-8010).

Point 2. To have good polydispersity index, the PDI should be below 0.3. However, the author showed the PDI higher than 0.4 (Figure 3A). Thus, the nanoparticle developed here doesn't have narrow size distribution.

Response: We appreciate the keen observation made by the reviewer. Yes, we agree with the reviewer. This has been addressed in the manuscript (Line 291).

Point 3. Since, the nanosponge has cytotoxic effect in the cancer model, the author should perform the release study in the acidic pH too.

Response: We do agree with the reviewer’s suggestion and carried out a release in an acidic medium (pH 4.5). For analysis, we have also plotted the standard curve in this media. The results have been incorporated into the manuscript (Lines 343-347, Figure 7B).

Point 4. Major in vitro experiment like intracellular uptake study of nanoparticle is missing.

Response: Yes, the suggestions by the reviewer are valuable and worth considering. As we have outsourced the cell line study, it is difficult for us to incorporate these suggestions in this work. However, we are working on skin cancer and will consider these for our future studies.

Point 5. The author should be consistent with the terminology B16F10, somewhere it's mentioned B16F12 as well.

Response: As per the reviewer’s suggestion, the terminology B16F12 has been thoroughly checked and made consistent throughout the manuscript.

Point 6. Animal experiment is encouraged to confirm its effect.

Response: We thank the reviewer for this suggestion. This work was a part of the Master's program project which is time-bound. However, considering the reviewer’s suggestion and the promising results obtained, we can take up animal studies in the future.

Round 2

Reviewer 1 Report

The Authors have responded sufficiently to some of the concerns metioned in the review, however some issues are still incorrect, and some were left without any appropriate comment. Below are the details:

1) the explanation added in the manuscript is correct

2) I can not see the figures showing absorbance spectrum and standard curve for sesamol. Even though they are correct, what about the standard curve for SES at pH 4.5?

3) the SEM images in the corrected manuscript are more informative, indeed

4) the Authors claim that "in the case of SES-NS, the release was found to be 60% in 24 hours (...), we have not continued release further". Why? It should be a basic information for SES-loaded NS characteristics, especially regarding that pH 7.4 resembles the physiological environment.

5) lines 362-363 - the denaturation was not reduced to 70.7% (!) Inhibition rate at 70 % means that denaturation was reduced by 70% or the Authors should explain what exactly do the mean with reduction in denaturation.

And there is no explanation on the link between inhibited denaturation and anti-inflammatory potential. How can the proteins denature under physiological conditions? How does the SES-NS protect them?

6) In my opinion we can not conclude about cytotoxicity against skin cancer cell lines (e.g. in the manuscript title) based only on one test performed on one cell line.

7) correction made by the Authors concerning the NS properties are acceptable

Author Response

Response to reviewer comments:

  1. the explanation added in the manuscript is correct.

Response: We are grateful to the reviewers for their insightful comments on our paper. We are thankful to the reviewers for this positive comment. We have further incorporated the new suggestions of the reviewer in the revised article.

  1. I can not see the figures showing absorbance spectrum and standard curve for sesamol. Even though they are correct, what about the standard curve for SES at pH 4.5?

Response: We are sorry for the confusion. The figures of absorbance spectra and standard curve of sesamol at pH 4.5 were included in the pdf version of the earlier response to reviewer comments, as we could not paste the images in the option provided. We have now again included the same in the pdf version of the responses uploaded.

  1. the SEM images in the corrected manuscript are more informative, indeed

Response: We thank the reviewer for this comment.

  1. the Authors claim that "in the case of SES-NS, the release was found to be 60% in 24 hours (...), we have not continued release further". Why? It should be a basic information for SES-loaded NS characteristics, especially regarding that pH 7.4 resembles the physiological environment.

Response: We are sorry for this. The release of SES in PBS (pH 7.4) has now been performed for 48 hours and results have been incorporated in the manuscript as suggested by the eminent reviewer.

  1. lines 362-363 - the denaturation was not reduced to 70.7% (!) Inhibition rate at 70 % means that denaturation was reduced by 70% or the Authors should explain what exactly do the mean with reduction in denaturation.

And there is no explanation on the link between inhibited denaturation and anti-inflammatory potential. How can the proteins denature under physiological conditions? How does the SES-NS protect them?

Response: We are sorry for the confusion. In lines 362-363, the suggestions by the reviewer have been incorporated. In addition, the explanation linking the denaturation and anti-inflammatory potential is also mentioned in Lines 366-369.

  1. In my opinion we can not conclude about cytotoxicity against skin cancer cell lines (e.g. in the manuscript title) based only on one test performed on one cell line.

Response: We thank the reviewer for this suggestion. The manuscript title has been modified according to the reviewer’s suggestion.

  1. correction made by the Authors concerning the NS properties are acceptable

Response: We thank the reviewer for this positive comment.

Round 3

Reviewer 1 Report

The Authors made substantial improvements to the previous version of manuscript and addressed all my comments. Please, improve the quality of Figure 7 (fonts and description of axes, because they are hardly legiblie).

Author Response

Response to comments:

  1. The Authors made substantial improvements to the previous version of manuscript and addressed all my comments. Please, improve the quality of Figure 7 (fonts and description of axes, because they are hardly legiblie).

Response: As suggested by the reviewer, we improved the quality of figure 7 and inserted in the manuscript.